

# Novel insights into the effect of drought stress on the development of root and caryopsis in barley

Fali Li[*], Xinyu Chen[*], Xurun Yu, Mingxin Chen, Wenyi Lu, Yunfei Wu and Fei Xiong

Jiangsu Key Laboratory of Crop Genetics and Physiology/Joint International Research Laboratory of Agriculture & Agri-Product Safety of the Ministry of Education/College of Biological Sciences and Technology, Yangzhou University, Yangzhou, China

[*] These authors contributed equally to this work.

## ABSTRACT

Drought is a common natural disaster in barley production, which restricts the growth and development of barley roots and caryopses seriously, thereby decreasing yield and debasing grain quality. However, mechanisms for how drought stress affects barley caryopses and roots development under drought stress are unclear. In this paper, Suluomai1 was treated with drought from flowering to caryopses mature stage. The morphological and structural changes in roots growth and caryopses development of barley were investigated. Drought stress increased root/shoot ratio and eventually led to the 20.16% reduction of ear weight and 7.75% reduction of 1,000-grain weight by affecting the biomass accumulation of roots and caryopses. The barley roots under drought had more lateral roots while the vessel number and volume of roots decreased. Meanwhile, drought stress accelerated the maturation of caryopses, resulting in a decrease in the accumulation of starch but a significant increase of protein accumulation in barley endosperm. There was a significantly positive correlation (0.76) between the area of root vessel and the relative area of protein in endosperm cells under normal condition and drought increased the correlation coefficient (0.81). Transcriptome analysis indicated that drought induced differential expressions of genes in caryopses were mainly involved in encoding storage proteins and protein synthesis pathways. In general, drought caused changes in the morphology and structure of barley roots, and the roots conveyed stress signals to caryopses, inducing differential expression of genes related to protein biosynthesis, ultimately leading to the increase in the accumulation of endosperm protein. The results not only deepen the study on drought mechanism of barley, but also provide theoretical basis for molecular breeding, high-yield cultivation and quality improvement in barley.

# INTRODUCTION

Barley (*Hordeum vulgare* L.) is an important food, feed and cash crop, and its planting area ranks fourth all around the world (*Lü, Wu & Fu, 2015*). Drought stress is a common natural disaster in agricultural production, which seriously restricts roots growth, caryopses

Corresponding author
Fei Xiong, feixiong@yzu.edu.cn

development and final yield of barley. Previous studies have found that barley roots can absorb inorganic salt and water and transport them to the aboveground parts (*Varney & Canny, 1993*; *Xiong et al., 2006*), so the roots morphological characteristics can be used as a key index for drought tolerance evaluation (*Chloupek et al., 2010*). Previous researches pointed that drought reduced the number of tillers, plant height and grains per ear of barley, resulting in a significant decrease in thousand kernel weight and yield of the ear (*Samarah, 2005*; *Samarah et al., 2009*). The accumulation of protein and starch was also affected by water deficit. Studies showed that protein content in grains increased, starch content and size changed, while starch structure did not change significantly under drought stress (*Nezhadahmadi, Prodhan & Faruq, 2013*; *Gous, Gilbert & Fox, 2015*; *Yu et al., 2017*). From the genetic level, scholars have confirmed that drought tolerance indices can be used as efficient criteria for screening drought-resistant and sensitive genotypes of barley (*Sharafi et al., 2015*). The discovery of drought-tolerant genes and their quantitative trait loci are of great significance to the breeding and quality improvement of barley (*Nevo & Chen, 2010*).

For plants, survival in adverse conditions needs substantial changes in the metabolism, which is reflected in extensive transcriptional level changes upon the occurrence of stress (*Janiak et al., 2018*). The transcriptome analysis can furnish with information about regulation of gene expression at transcriptional levels and provide an insight into the mechanisms underlying stress responses. Since the draft genome of barley (http://plants.ensembl.org/Hordeum_vulgare/Info/Index) has been available for years, researches on transcriptome analysis of barley have increased rapidly. In previous studies, the root hair morphology and transcriptional characteristics of two contrasting Tibetan wild barley genotypes and drought-tolerant cultivar were investigated and then the full length cDNA of a novel $\beta$-expansin gene (*HvEXPB7*) was cloned, which is the unique root hair development related gene (*He et al., 2015*; *Kokáš, Vojta & Galuszka, 2016*). Additionally, *Abebe et al. (2010)* compared response of the transcriptome of the lemma, pale, awn, and seed to drought and found that transcript abundance followed the water status of the spike organs, while *Vojta et al. (2016)* conducted a detailed transcriptomic analysis on leaves of transgenic plants subjected to re-watering after drought stress. The results revealed that the up-regulated expression of genes encoding putative enzymes involved in production of jasmonates and other volatile compounds caused a faster tendency of return to initial photochemical activities compared to wild-type.

The effects of drought on the growth of barley reflect on the underground and aboveground organs. Previous studies about drought effect of barley have focused on roots morphologies and yield traits (*Barnabás, Jäger & Fehér, 2008*; *Afsharibehbahanizadeh et al., 2014*; *Tyagi et al., 2014*; *Haddadin, 2015*; *Hannah et al., 2018*). However, few studies have been reported on roots structure characteristics and caryopses development in barley under drought, and their relationships are unclear. In the present study, Suluomai1 (SLM1) was subjected to drought stress from flowering to caryopses mature stage, and the morphological and microstructural changes in roots and caryopses were observed. Meanwhile, transcriptome analysis was carried out to investigate the possible mechanism underlying the barley caryopses development responding to drought. The results may

provide valuable information for revealing the relationship between barley roots and caryopses development under drought, and lay a theoretical foundation for high-yield cultivation and quality improvement in barley.

## MATERIALS & METHODS

### Plant materials and Drought treatment

The barley variety selected in this study was SLM1, which was provided by the Agricultural College of Yangzhou University, Jiangsu Province, China. Seeds were sown in plastic pots (30 cm × 30 cm, 10 seeds per pot), which were placed in rainproof shelters under drought simulation in the experimental field of Key Laboratory of Crop Genetics and Physiology in Yangzhou University from September 2017 to May 2018. The soil was sandy loam containing organic matter (2.45%), available nitrogen (106 mg/kg), available phosphorus (33.8 mg/kg), and available potassium (66.4 mg/kg). Plants were thinned to eight plants per pot 2 weeks after sowing. A minupressure soil hygrometer (SP-11, Institute of Soil Science in Nanjing, China) was inserted into the soil at a depth of 15 cm to detect the soil water potential. Barley plants were accurately irrigated from flowering to caryopses mature stage to maintain the water potential at $-20$ and $-60$ kPa, which reflected the optimum level of control condition (CC) and drought stress (DS), respectively. Each treatment contained 30 pots and plants were selected from different pot as the replication. During barley flowering stage, the flowering ears were tagged to label the anthesis date, and the roots and caryopses at different days post anthesis (DPA) were collected and tested.

### Roots and caryopses morphology observation and growth indices determination

Barley roots and caryopses samples were collected at 10, 20 and 30 DPA, and their morphologies, including shapes, colors and so on, were observed and photographed. The fresh weight of roots, canopy structure and 1,000-grain of caryopses were conducted. Then, the samples were placed in a fan-forced oven at 105 °C for 1 h, and then baked at 42 °C to attain constant weights for dry weight determination. The root/shoot ratio and water content were calculated. Meanwhile, the ripe barley ears were collected to measure the indices such as ear length, ear weight, grain number per ear and 1,000-grain weight. The protein content of mature caryopses was measured by kjeldahl nitrogen determination method.

### Microstructure observation of roots and caryopses

Barely caryopses and roots at 10, 20 and 30 DPA were acquired. The caryopses selected were in the same position of the middle of main ear and the roots samples were the segments of secondary root which were 2 cm from the stem axis base. The samples were cut transversely into 2 mm slices from the middle using a razor blade. The slices were soaked in 2.5% glutaraldehyde fixative [25% glutaraldehyde diluted 10 times at pH 7.2 phosphate buffered solution (PBS)] at 4 °C for 48 h immediately. The fixed samples were subsequently rinsed thrice with PBS and dehydrated in a graded ethanol series [20, 40, 60, 80, 90, 95, and 100% (thrice)], followed by propylene oxide replacement. Afterward, the samples were

infiltrated and embedded in low-viscosity Spurr's resin and polymerized at 70 °C for 12 h. The samples were cut into 1 μm slices using an ultramicrotome (Ultracut R, Leica, Germany) and pasted onto glass slides. Then, the slices were stained with 0.5% methyl violet or toluidine blue, rinsed, dried, and observed under a light microscope (DMLS, Leica, Germany). Photographs were captured using a CCD camera (Truechrome II, Truechrome, China) attached to the light microscope. Each treatment contained three replicated samples and each roots and caryopses sample was from different plants of different pots.

## Structural characteristics analysis of roots and caryopses

Image-Pro Plus (ver. 6.0, Media Cybernetics, USA) and Photoshop (ver. CC 2017, Adobe, USA) were used to analyze structural characteristics of roots and caryopses based on microphotographs as previously described (*Yu et al., 2015*). The roots were photographed at 100 times magnification and the transversal section areas of root vessel were measured. Meanwhile, two main endosperm regions, including abdominal and dorsal endosperm were photographed at 200 times magnification. The number of starch granules in endosperm cells was counted. Also, the areas of starch granules, protein bodies and their corresponding endosperm cells were measured. The ratios of starch granule and protein body areas to corresponding endosperm cell areas were calculated, defining as the relative areas of starch granules and protein bodies respectively. Each treatment conducted the resin slicing of three roots and caryopses samples from different plants and each sample selected ten microphotographs for the analysis.

## RNA extraction and sequencing

Barely caryopses from the middle of main ear were collected at 10 DPA, which was the beginning stage of grain filling. Total RNA was extracted using TRIzol reagent (Invitrogen, USA) following the manufacturer's protocol. Each treatment conducted three replicates of RNA samples for the cDNA library construction. Genomic DNA in the samples was removed using RNase-free DNase (Promega, USA). The mRNA samples were enriched by using oligo (dT)-magnetic beads and then cut into fragments with fragmentation buffer to meet the requirement of sequencing (long fragments may reduce the accuracy of sequencing). The cDNA was synthesized by reverse transcription using the fragments as templates and added with double-stranded DNA poly A and adaptor sequences after purification and terminal repair. Subsequently, cDNA libraries were constructed by polymerase chain reaction (PCR) amplification after selecting for fragment size and undergoing a quality check with an Agilent 2100 Bioanalyzer system. Finally, four qualified libraries (two libraries for each treatment) were sequenced with an Illumina HiSeq 2500 system by OE Biotech Co., Ltd. (Shanghai, China).

## Bioinformatics analysis of RNA-Seq

Clean reads were obtained from the raw reads after filtering out low-quality reads (quality threshold <20, length threshold <35 bp). The clean reads were blast to the *H. vulgare* cDNA database from Ensembl (ftp://ftp.ensemblgenomes.org/pub/plants/release-44/fasta/hordeum_vulgare/cdna/Hordeum_vulgare.IBSC_v2.cdna.all.fa.gz) using Tophat/bowtie2 software. Transcript expression was carried out and differentially expressed genes (DEGs)

were screened using a 2-fold change at the $P < 0.05$ level (*Trapnell et al., 2010*). The functional annotation of DEGs was performed using Gene Ontology (GO) and Kyoto Encyclopedia of Genes and Genomes (KEGG) pathway analysis.

### Statistical analysis

Statistical analysis was performed using Microsoft Excel (ver. 2016, Microsoft Corp., USA) and SPSS (ver. 19.0, SPSS Inc., USA). The significant analysis was conducted using $t$-test at a probability significance level of $P < 0.05$ according three replicates. Figure production was using Photoshop and Origin (ver. 2017, Origin Lab., USA).

## RESULTS

### Changes in roots growth and caryopses development of barley under DS

Under DS, the fresh and dry weight of barley roots increased first and then decreased, which was significantly higher than that under CC at 10 and 20 DPA (Figs. 1A and 1B). However, the water content of roots had a tendency of decrease as the growth of roots under both CC and DS. The water content of roots under DS was significantly lower than that under CC at 10 and 20 DPA (Fig. 1C). The fresh weight of caryopses increased first and then decreased but the dry weight of caryopses continually increased during the development of caryopses under CC and DS. Before 20 DPA, the fresh and dry weight of caryopses under DS were significantly higher than that under CC. But DS decreased the fresh and dry weight of caryopses at 30 DPA (Figs. 1D and 1E). The water content decreased with caryopses development and fluctuated around the control level when treated with drought (Fig. 1F). Additionally, DS significantly increased the root/shoot ratio during whole development stage of barley (Fig. 1G). Moreover, DS not only affected the biomass accumulation of plant but also resulted in the changes of ear characteristics with shorter ear length, smaller number of grains per ear, lower ear weight and 1,000-grian weight (Table 1). Meanwhile, DS significantly increased the content of protein in mature caryopses.

### Changes in morphology and structure of barley roots under DS

The morphology of barley roots changed when subjected to DS. The barley had a longer, larger and more complex roots system with more lateral roots under DS, compared with control (Figs. 2A and 2B). The small segments of barley secondary root which were 2 cm from the stem axis base were selected for microstructure observation and the cortex and xylem vessel were observed. As the growth of roots, the number of vessels increased under CC while decreased under DS (Figs. 2C–2H). This indicated that DS decreased the vessel numbers in barley roots. Meanwhile, DS also significantly reduced the area of root vessel transversal section at 10 and 20 DPA according to the statistical data (Fig. 2I).

### Changes in morphology and structure of barley caryopses under DS

The morphology and microstructure of barley caryopses at 10, 20 and 30 DPA were observed by resin semi-thin slicing, and the number of starch granules along with relative areas of starch granules and protein bodies in endosperm was also counted using Image-Pro Plus software (Fig. 3). During the development of barley caryopses, DS affected caryopses
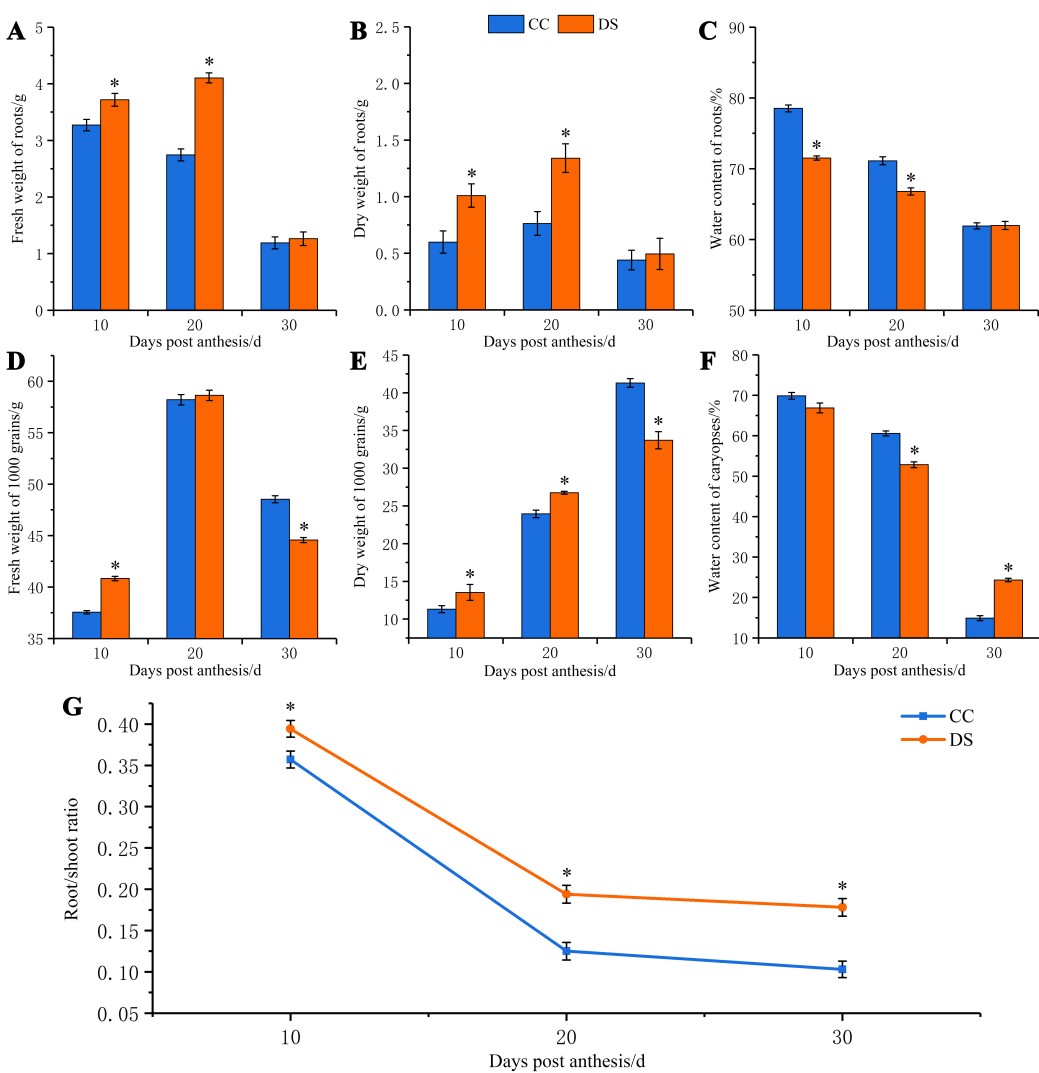

**Figure 1** **Changes in fresh and dry weight of roots and caryopses and root/shoot ratio in barley under DS.** (A) Fresh weight of roots. (B) Dry weight of roots. (C) Water content of roots. (D) Fresh weight of caryopses. (E) Dry weight of caryopses. (F) Water content of caryopses. (G) Root/shoot ratio. Error bars on the graphs indicate the standard deviation calculated from three replications. Asterisks indicate significant difference between treatments at $p < 0.05$ as determined by $t$-test. CC, control condition; DS, drought stress.

**Table 1** **Changes in ear characteristics and caryopses protein content of mature barley under DS.** Data are shown as means $\pm$ SD, $n = 3$. For each column, asterisks indicate significant difference between treatments at $p < 0.05$ ($t$-test).

| Ear characteristics | Ear length (cm) | Ear weight (g) | Number of grains per ear (No.) | 1,000-grain weight (g) | Protein content of mature caryopses |
|---|---|---|---|---|---|
| CC | 5.75 ± 0.13 | 1.31 ± 0.07 | 22.00 ± 1.00 | 46.44 ± 0.81 | 10.36 ± 0.59 |
| DS | 5.33 ± 0.20* | 1.04 ± 0.04* | 19.83 ± 0.95 | 42.84 ± 0.18* | 12.04 ± 0.73* |

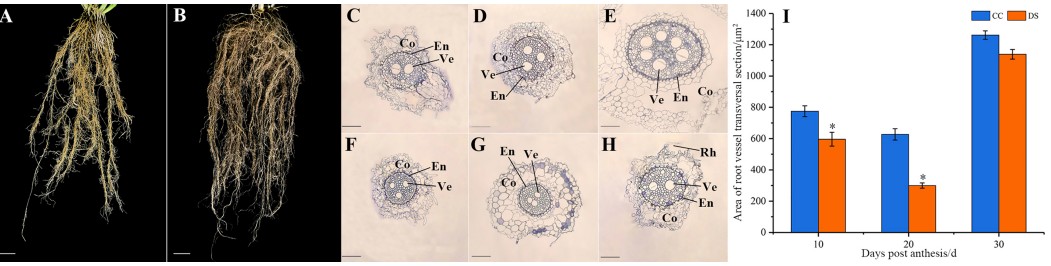

**Figure 2  Changes in roots morphology and structure of barley under DS.** (A, B) Morphology of roots under CC and DS. (C–E) Microstructure of root transversal section at 10, 20 and 30 DPA under CC. (F–H) Microstructure of root transversal section at 10, 20 and 30 DPA under DS. (I) Area of root vessel transversal section. Error bars on the graphs indicate the standard deviation calculated from three replications. Asterisks above the histograms indicate significant difference between treatments at $p < 0.05$ as determined by $t$-test. CC, control condition; Co, cortex; DS, drought stress; RH, root hair; En, endodermis; Ve, vessel. Scale bars: (A, B) 20 mm, (C–H) 100 μm.

morphology and endosperm substance accumulation. When barley was treated with DS, the epidermis of caryopses turned to yellow and was shrunken at the earlier stage compared to CC (Fig. 3A). This indicated that DS promoted the early maturity of caryopses. The abdominal endosperm is located on either side of the caryopses crease region, whereas the dorsal endosperm region is located facing the caryopses crease region. At 10 DPA, a number of starch granules and protein bodies were observed in dorsal and abdominal endosperm under DS while few were observed under CC (Figs. 3B–3E). The statistical data showed that DS significantly increased the relative areas of protein bodies in dorsal endosperm and the relative areas and number of starch granules in both dorsal and abdominal endosperm (Figs. 3N–3P). At 20 DPA, the number and volume of starch granules in endosperm increased along with the accumulation of protein bodies under both CC and DS and some small starch granules began to occur (Figs. 3F–3H). However, several large protein aggregations consisting of many small protein bodies units assembled in abdominal endosperm under DS (Fig. 3I). Meanwhile, there was no significant difference in the accumulation of starch granules in dorsal and abdominal endosperm between CC and DS but a significant increase of protein bodies accumulation in abdominal endosperm under DS (Figs. 3N–3P). At 30 DPA, endosperm cells were almost occupied by starch. Starch granules squeezed to deformed and protein bodies filled into the gap between starch granules. More small starch granules in endosperm under DS were observed compared to CC, but the number and relative areas of starch granules in endosperm under DS were lower than those under CC (Figs. 3J–3M, 3O–3P). Additionally, DS significantly increased the relative areas of protein bodies in dorsal and abdominal endosperm (Fig. 3N).

According to the results above, it can be concluded that DS promoted the precocity and shortened the development process of caryopses. Moreover, DS affected the substance accumulation of endosperm, which showed an increase in protein accumulation and a decrease in final starch accumulation.
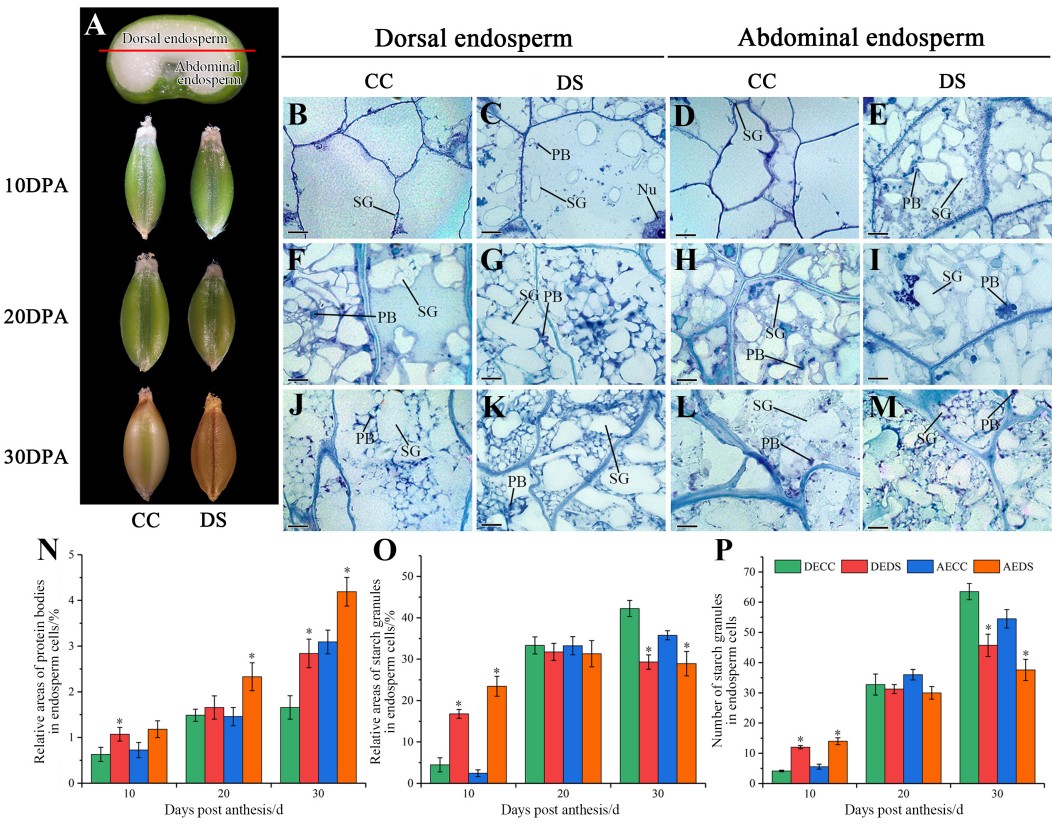

**Figure 3** **Changes in caryopses morphology and structure of barley under DS.** (A) Division of abdominal and dorsal endosperm and morphology of caryopses. (B–M) Microstructure of endosperm under CC and DS at 10, 20, 30 DPA. (N) Relative areas of protein bodies in endosperm cells. (O) Relative areas of starch granules in endosperm cells. (P) Number of starch granules in endosperm cells. Error bars on the graphs indicate the standard deviation calculated from three replications. Asterisks above the histograms indicate significant difference between treatments at $p < 0.05$ as determined by $t$-test. AECC, abdominal endosperm under control condition; AEDS, abdominal endosperm under drought stress; DECC, dorsal endosperm under control condition; DEDS, dorsal endosperm under drought stress; Nu, nucleus; PB, protein bodies; SG, starch granules. Scale bars: (B–M) 50 μm.

## Correlation analysis of roots and caryopses structure under DS

To investigate the relationship between barley roots and endosperm substance accumulation under DS, the correlation analysis on the structure of roots and caryopses was conducted. The correlation coefficient between the area of root vessel transversal section and the relative areas of starch granules in endosperm cells was 0.44, indicating moderate correlation, and DS reduced its correlation coefficient to 0.36. However, the area of root vessel transversal section and the relative areas of protein bodies in endosperm cells had a significant strong correlation (0.76) and DS increased the correlation coefficient to 0.81, whose difference was significant. Therefore, the area of root vessel transversal section was more closely correlated with the relative areas of protein bodies compared to the relative areas of starch granules, and the correlation increased after treating with drought. The

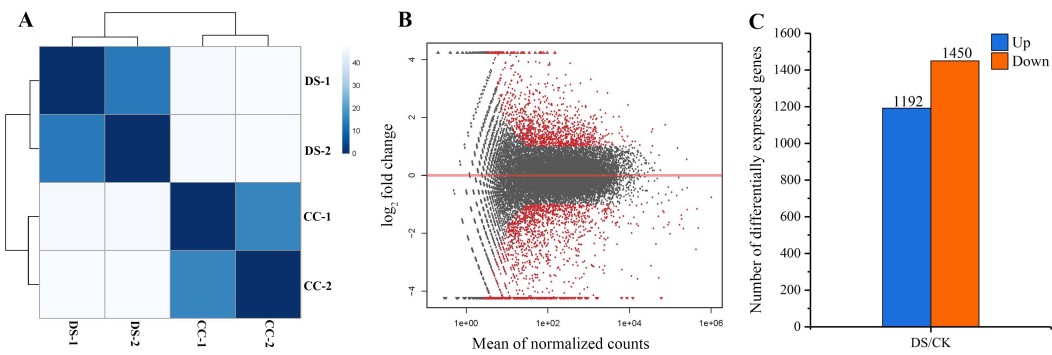

**Figure 4** **Analysis of gene differential expression.** (A) Sample to sample clustering analysis. (B) MA diagram of gene differential expression. (C) Number of up-regulated and down-regulated genes in barley caryopses under DS. Red dots denote DEGs with fold change > 2 and $p < 0.05$ in (B). The total numbers of up-regulated and down-regulated genes are displayed above the bars in (C).

results indicated that changes of roots structure under DS had a greater influence on the accumulation of endosperm protein bodies.

## Analysis of gene differential expression

In total, 258 billion clean reads were obtained in four samples after filtering, with approximate 64 billion reads on average from each sample. The results of sample to sample clustering analysis showed that the distance between CC and DS treatments was long while the similarity of gene expression pattern in two replicates was high (Fig. 4A). This indicated that the repeatability of samples in this study was reliable. DEGs were screened out at the threshold of 2-fold change as a basis of $P < 0.05$ (Fig. 4B). A total of 2,642 DEGs in barley caryopses were identified between CC and DS. Among these DEGs, 1,192 genes were up-regulated and 1,450 genes were down-regulated (Fig. 4C).

## DEGs involved in caryopses storage protein synthesis

The information of functional annotations for all DEGs was obtained from various databases, including Non-Redundant protein (Nr), Swissprot, GO and KEGG database. A total of 30 DEGs involved in encoding caryopses storage protein were screened out and specific information was shown in Table 2. Among these DEGs, nine of them showed up-regulated and 21 showed down-regulated expression under DS. Specifically, three genes encoding gliadin and nine genes encoding glutenin showed down-regulated expression under DS. Among DEGs encoding seed storage 2S albumin, eight genes were up-regulated and seven were down-regulated. Meanwhile, there were three genes encoding 11S seed storage protein, one of which expressed with up regulation and two showed down-regulated expression.

To gain more insights about DEGs, GO enrichment and KEGG pathway analysis were also implemented. Functional annotation based on GO and KEGG databases revealed that DEGs had various functions in biological process, cellular component and molecular function. To investigate the effects of DS on protein synthesis in barley caryopses, 14 GO terms that might be involved in protein biosynthesis were selected, including protein

**Table 2** Functional annotation of DEGs encoding caryopses storage protein.

| Gene ID | Description | Up/Down |
|---|---|---|
| HORVU1Hr1G000540 | Omega-gliadin | Down |
| HORVU1Hr1G000640 | Gamma-gliadin | Down |
| HORVU1Hr1G005150 | Hor1-17 C-hordein | Down |
| HORVU1Hr1G064080, HORVU1Hr1G066650 | Glutenin, high molecular weight subunit 12 | Down |
| HORVU1Hr1G000990, HORVU1Hr1G001020, HORVU1Hr1G001080, HORVU1Hr1G001140, HORVU1Hr1G001350, HORVU1Hr1G001420, HORVU1Hr1G005460 | Low molecular weight glutenin subunit | Down |
| HORVU1Hr1G008130 | 11S seed storage protein | Up |
| HORVU3Hr1G116010, HORVU7Hr1G119320 | 11S seed storage protein | Down |
| HORVU4Hr1G008420, HORVU4Hr1G071790, HORVU7Hr1G105910, HORVU7Hr1G106040, HORVU7Hr1G106130, HORVU0Hr1G022170, HORVU2Hr1G080570, HORVU2Hr1G099450 | Seed storage 2S albumin superfamily protein | Up |
| HORVU4Hr1G074480, HORVU4Hr1G087780, HORVU5Hr1G109240, HORVU7Hr1G027440, HORVU7Hr1G115810, HORVU7Hr1G115830, HORVU1Hr1G054040 | Seed storage 2S albumin superfamily protein | Down |

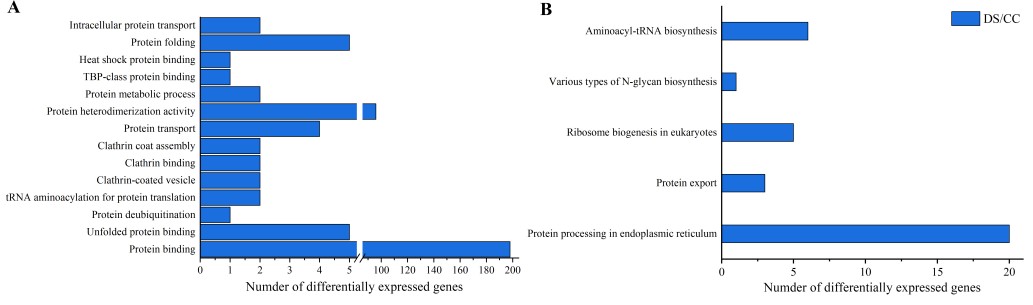

**Figure 5  GO enrichment and KEGG pathway analysis of DEGs.** (A) Number of DEGs in GO terms involved in protein biosynthesis. (B) Number of DEGs in the KEGG pathway involved in protein biosynthesis.

folding, protein binding, protein transport, intracellular protein transport and so on. Among them, the DEGs annotated to protein binding entry was the most (198 DEGs), followed by protein heterodimerization activity entry (96 DEGs) (Fig. 5A). Similarly, five possible KEGG pathways related to protein biosynthesis were also screened. They were protein processing in endoplasmic reticulum (ER), protein export, ribosome biogenesis in eukaryotes, various types of N-glycan biosynthesis, aminoacyl-tRNA biosynthesis. The number of DEGs enriched in the pathway of protein processing in ER was the largest and six DEGs were enriched in aminoacyl-tRNA biosynthesis pathway, which took the second place (Fig. 5B).

In order to further investigate the possible mechanism underlying the protein biosynthesis in barley caryopses under DS, the pathway of protein processing in ER

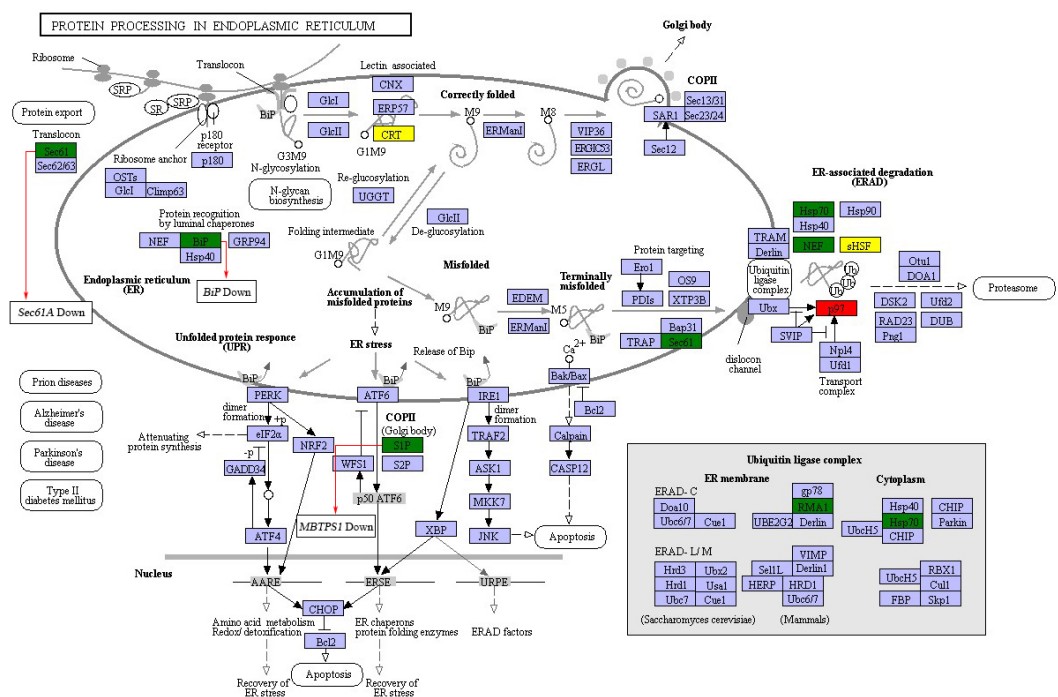

**Figure 6** **DEGs in protein processing in ER pathway.** Red and green represent proteins encoded by up-regulated and down-regulated genes, respectively. Yellow represent proteins encoded by both up-regulated and down-regulated genes.

was chosen, which is the key process in the formation of endosperm protein bodies. The up-regulated and down-regulated DEGs enriched in the pathway were shown in Fig. 6. The process of protein synthesis and sorting in ER has been revealed in previous literatures (*Crofts et al., 2005*; *Määttänena et al., 2010*). The newly synthesized peptide chain entered the ER through the translocation of the Sec61 pore, followed by N-glycosylation. The correctly folded proteins were packaged into the vesicle II (COPII) transport vesicles, which were then transported into the Golgi apparatus. The unfolded or misfolded proteins were remained in the ER and eventually entered the ubiquitin-mediated proteasome degradation process. During this process, the expression of certain genes altered under DS. For example, *SEC61A* and *BIP* showed the expression of down regulation, which both encoded protein transport protein subunit. *MBTPS1* encoded membrane-bound transcription factor site-1 protease in COPII and was down-regulated under DS. The results indicated that DS altered the expression of some certain genes during the process of protein export and COPII transport vesicle formation in protein processing in ER pathway. However, the specific mechanism of these genes in regulation of endosperm protein accumulation in response to DS needs further verification.

## DISCUSSION

The growth of roots is related to plant growth stage, genotype, arbuscular mycorrhizal colonization rate in soil and can be affected by environmental factors such as drought

(*Akman & Topal, 2016*; *Sendek et al., 2019*). Generally, the roots of plants firstly sensed the stress signal when subjected to DS. Then the morphology and structure of roots changed to help the plant to absorb water more efficiently, thus adapting to drought. The vessel in root xylem is the channel for water and inorganic salt transportation, which is crucial for coping with water deficit. In this study, DS reduced the area of root vessel transversal section in barley, which can be explained from the aspect of water flow conductivity. The water flow conductance of roots is affected by radial and axial resistance. When the diameter of vessel is too large, the bubbles in the vessel will form embolism more easily, which will increase the radial and axial hydraulic resistance and reduce the root water flow conductivity. Therefore, the root water absorption is limited and it is not conducive to adapt to drought for plant growth (*Thorsten & Wieland, 2011*; *Vadez, 2014*; *Bartlett et al., 2016*; *Niu et al., 2016*). While, the narrower xylem vessels could overcome this phenomenon, which was consistent with the results in wheat (*Comas et al., 2013*).

Barley caryopsis is an important organ for nutrient storage and its development is also limited by DS. Previous studies found that drought caused the early senescence of plants and affected the grain filling by shortening grain filling time and reducing grain filling rate (*Shi et al., 2016*). This was consistent with our study, which found the precocity of barley caryopses under DS. Meanwhile, DS significantly decreased the accumulation of starch granules but increased the accumulation of protein bodies in endosperm of barley caryopses. Transcriptome analysis showed that DS induced differential expression of genes were mainly related to storage proteins and protein synthesis pathways, thus regulating protein biosynthesis and eventually leading to the increase of protein accumulation in barley caryopses.

As the important organ for water and nutrient absorption in barley, the morphological and structural characteristics of roots are closely related to the development of caryopses (*Ramireddy et al., 2018*). In this study, DS caused changes in roots and caryopses weight and increased root/shoot ratio, which eventually led to a decrease in the ear weight and 1,000-grain weight. Similar results were also reported in previous papers (*Samarah, 2005*; *Samarah et al., 2009*). Additionally, the correlation analysis between roots structural characteristics and endosperm substance accumulation was conducted under CC and DS. It was found that the area of root vessel had a strong correlation with the area of protein bodies in endosperm cells and drought increased the correlation coefficient. This indicated that the changes of roots structure caused by DS might have a greater influence on the accumulation of endosperm protein compared to endosperm starch accumulation. In general, the morphology and structure of barley roots changed under DS and the roots transported stress signals to caryopses. The drought signal induced differential expression of genes related to protein biosynthesis in caryopses and ultimately resulted in the increase of endosperm protein accumulation.

## CONCLUSIONS

In this study, the morphological and structural changes in roots growth and caryopses development of barley under DS were investigated. Under DS, the roots of barley had

more lateral roots with a narrower vessel structure. Meanwhile, DS promoted the precocity of caryopses and affected the substance accumulation in caryopses with a decrease in endosperm starch but an increase in endosperm protein. DS induced changes in roots and caryopses weight led to the increase of root/shoot ratio but a decrease in 1,000-grain weight. While, the changes of roots structure caused by DS had a greater influence on the accumulation of endosperm protein. Transcriptome analysis indicated that DS induced differential expression of genes were mainly related to protein biosynthesis in caryopses. In conclusion, the roots of barley firstly sensed the stress signals when subjected to DS, which led to changes in morphology and structure of roots and substance accumulation of caryopses. At the same time, the stress signals from roots to caryopses induced the differential expression of genes involved in encoding storage proteins and protein biosynthesis pathways, ultimately leading to the increase in endosperm protein accumulation in barley caryopses. These results can provide novel insight into the drought-related researches in barley.

### Funding
This work was supported by the Natural Science Foundation of Jiangsu Province (BK20170497), the China Postdoctoral Science Foundation (2018M642332) and the Practical Innovation Training Plan for College Students in Jiangsu Province (201811117003Z). The funders had no role in study design, data collection and analysis, decision to publish, or preparation of the manuscript.

### Grant Disclosures
The following grant information was disclosed by the authors:
Natural Science Foundation of Jiangsu Province: BK20170497.
China Postdoctoral Science Foundation: 2018M642332.
Practical Innovation Training Plan for College Students in Jiangsu Province: 201811117003Z.

### Competing Interests
The authors declare there are no competing interests.

### Author Contributions
- Fali Li and Xinyu Chen performed the experiments, analyzed the data, prepared figures and/or tables, authored or reviewed drafts of the paper, and approved the final draft.
- Xurun Yu analyzed the data, authored or reviewed drafts of the paper, and approved the final draft.
- Mingxin Chen and Wenyi Lu performed the experiments, prepared figures and/or tables, and approved the final draft.
- Yunfei Wu and Fei Xiong conceived and designed the experiments, authored or reviewed drafts of the paper, and approved the final draft.

## Data Availability

The raw measurements are available in the Supplemental File.

## Supplemental Information

Supplemental information for this article can be found online at http://dx.doi.org/10.7717/peerj.8469#supplemental-information.

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
