# Peer review of "Novel insights into the effect of drought stress on the development of root and caryopsis in barley"

_PeerJ, doi:10.7717/peerj.8469_

## Round 0.1 · original submission · Major Revisions

We have received comments from 4 reviewers, most of them recommended for minor revision, however R3 has some serious concerns which you must address adequately. Please take all comments into consideration and make revision accordingly.

Reviewer 1 ·

Basic reporting

In the present study, barley was subjected to drought stress and the morphology and microstructural changes in root and caryopsis were observed. The pictures in the microscope seem beautiful and clear. At the same time transcriptome analysis was carried out to investigate the mechanism underlying the barley caryopsis development responding to drought stress. The results may provide valuable information for revealing the barley root and caryopsis development under drought stress. The data are enough to lead their conclusion, and should give new criteria of barley development under drought stress. Therefore, in my opinion, the paper is acceptable for publication with some revision.

Experimental design

.

Validity of the findings

.

Additional comments

1. The abstract of the manuscript does not seem to be as informative as it should be. For example, the variety of barley used for the experiment and drought stress imposition stage should be mentioned in the abstract. Also the specific value of changes in barley traits under drought stress may be mentioned in abstract to make it more informative rather than generalized.
2. Line 91-92, the unit of soil nitrogen, phosphorus and potassium content seems incorrect. Please check again.
3. Line 107, the description of 1000-grain weight should keep consistent in the context.
4. There is a lack of the scale bar in Figure 2B.
5. There are some writing mistakes needing to be improved in the manuscript. Please check it carefully.

·

Basic reporting

No comment

Experimental design

No comment

Validity of the findings

No comment

Additional comments

This manuscript descripted how drought stress affects barley root growth and caryopsis development from morphological structure changes to molecular work (gene expression). The authors found drought stress increased lateral roots and decreased the vessel number and volume of root. Drought stress accelerated the maturation of caryopsis with less starch accumulation but more protein accumulation in barley endosperm. Transcriptome analysis found that differential expressions of genes in caryopsis involved in encoding storage proteins and protein synthesis pathways. These findings provide some fundamental knowledge for drought stress, which is an increasing problem in agricultural production.
However, I have some questions.
1. How much protein contents change in caryopsis by drought stress?
2. How to determine grain quality? More proteins is better?
3. Lines 254-266 need to be improved.
4. Some figures need to be improved (Fig. 1A, B, Fig. 3N,O,P, Fig. 6)
Overall, the manuscript is good. But, it still needs improving in writing (see more details in attached file).

Reviewer 3 ·

Basic reporting

This manuscript described an experiment in which a post-anthesis water stress was applied to one barley cultivar in a single experimental replicate under glasshouse conditions. Using light microscopy, the structure of root xylem was analysed, and starch granules and protein bodies in the caryopsis endosperm were quantified by area. RNA-seq followed by a gene ontology analysis were used to detect changes in trascript abundance between the water-stressed plants and the well-irrigated plants. The introduction covered the necessary literature to understand the objectives of the study. Unfortunately the quality of the English throughout the manuscript means that some of the meaning is ambiguous, and there are a large number of grammatical errors throughout the text adding to the difficulty in interpretation. Some of these are highlighted in the General Comments section, although they are too numerous to mention every one. The findings reported are typical of water stress applied during grain filling, and the authors are advised to make it clear how their results add to this knowledge.

Experimental design

I have some serious concerns regarding the experimental design, and am not convinced that these are able to be rectified in order to make the results meaningfully interpretable.

The research question seemed to be aimed at correlating water-stressed changes in root development with development of the barley caryopses under these conditions. This is an interesting question, and had the experimental design been reported more accurately, the results could be interpreted as a meaningful correlation. However, there was no indication of any true replication in this experiment, which appeared to consist of one experiment containing 30 pots. It was not indicated that the pots were grown either spatially or temporally separated in a way that could be thought of as replication, and no indication that the caryopses and roots were taken from pots with meaningful replication in mind. Furthermore, the authors should state where the barley caryopses and roots were sampled from in order to minimise experimental bias. For example, were the caryopses always harvested from the main shoots, and at the same position within the ear ? Were the roots harvested from the main root axis, and how long from either the stem axis or the root tip ? It is therefore impossible to determine whether the correlations observed are genuine, or due to sampling bias. Why was the biomass not reported ? The root:shoot ratio is ambiguous if both were reduced by the water stress treatment.

There was a secondary emphasis on using RNA-seq to determine a genetic mechanism for these correlations. I have concerns regarding the experimental design to address these questions. The authors acknowledge that the development rate of the caryopses were affected by the water stress conditions, which means that using RNA-seq to compare caryopses at the same number of days post-anthesis would result in comparison of caryopses at two different growth stages. Therefore the reported results should be treated with caution, as the changes in transcript abundance in barley caryopses among even close growth stages will conflate these results, and would include exactly the gene types reported here (seed storage proteins for example). In addition to the lack of meaningful replication discussed above, the validity of these results are difficult to judge.

Validity of the findings

The validity of the findings is difficult to interpret as discussed above in the section for experimental design. The design does not allow for much in the way of meaningful interpretation.

Additional comments

The English quality should be improved to allow better interpretation of the meaning. There are some errors throughout the text. I suggest having the manuscript corrected for English throughout, paying particular attention to the use of plurals, and ambiguous text.
I give some examples for the abstract and the beginning of the introduction, but to address every instance would require more than a reviewer.
Abstract
Line 32 (and throughout), the plural of caryopsis is « caryopses »
Line 33 « thus ultimately led to » should be either « this ultimately led to » or « ultimately leading to ».
Lines 34 to 36, it is not made clear what these results have to do with molecular breeding, or yield and quality improvement.

Introduction
Line 42 « roots » of barley
Line 45 « Seniors pointed out », I assume Seniors is an author but this study is not referenced
Lines 46 and 47 barley does not have panicles, I suggest reading a basic cereal anatomy textbook for this
Line 52 « indices » is the plural of index, the plural of criterion is « criteria »
Line 55 « plant » and « need » should be plural
Line 59 the barley genome sources should be properly cited

Materials and methods
Lines 91 and 92 there is am odd symbol included for the soil compositions reported
Line 97 there should be some indication of how/if replication was determined. 30 pots grown at the same time is not meaningful replication.
Lines 97 to 99 please clarify what you mean by different development stage
Line 102 define what is meant by morphology
Line 104 and throughout, there should be no space between the numerals and the units for °C
Line 109 further information is needed about barley caropses harvest (see above comments on experimental design and validity). Pay particular attention to how replication was determined, which shoots, from how many pots, how was the replication biologically meaningful ?
Line 125 and throughout, I don’t know what is meant be « abdominal », perhaps « ventral ». I suggest a cereal anatomy textbook.
Line 129 Define what is a replicate
Line 132, again be very precise about how the caryopses were harvested, and define their growth stages, because otherwise these results cannot be meaningfully interpreted.
Line 152 The statistical analysis sections requires a lot more information in order to determine which tests were applied to which results. How were the results corrected for multiple comparisons ? Or were results only compared to one « control » results ? This needs to be well defined. The statistical analyses may need to be re-done according to the number of replicates, because 30 pots in one experiment does not necessarily mean 30 replicates.

Results
The p-values need to be reported throughout, because it is not always clear what is meant by « significant ».
The error bars should be clearly explained on all graphs, paying particular attendtion to stating the true number of replications. If there was no replication the errors should be standard deviation.
Line 163 « and DS » is repeated
Lines 174 to 176 Are there any analyses to support this ? If this is true, it makes it even more important to specify exactly where and how anatomically the sections were taken for microscopic analysis.
Lines 187 to 188 That water stress resulted in early caryopsis maturation is typical of water stress during grain filling, and makes it particularly important that the authors specify the growth stages (not days post anthesis) at which caryopses were harvested, otherwise readers will have to assume that the results cannot be compared between treatments as the caryopses could be at significantly different growth stages.
Lines 209 In this section, the p-values should be given alongside the correlations, as the correlation alone does not indicate significance.
Line 231 « various database(s) » could be more specific
Line 236 is « albumin » really what the authors mean ?
Line 240 to 241 is very vague
Line 252 justify why this particular pathway was chosen, what meaning does it specifically have to the research question ?
Lines 286 to 289 is very speculative, particularly regarding that the RNA-seq results were likely from caryopses at different growth stages.
Lines 298 to 300 this is very speculative and the study does not contain sufficient information to draw any causative meaning between observed correlations.
Line 316 the study present no evidence of water and nutrient absorption being affected, and this statement should be removed.
Line 320 to 321 I am not convinced the results are novel, the authors could spend furhter time addressing how their study fills gaps in the literature.

Reviewer 4 ·

Basic reporting

Li et at. al describe in their paper the morphological and structural changes in root growth and caryopsis development of barley under drought stress. Transcriptional analyses showed that drought stress induce differential expression of genes that are mainly related to protein biosynthesis in caryopsis. The authors finally suggest that the root of barley firstly senses the stress signal when subjected to drought stress which led to changes in root morphology and structure. Subsequently, the water and nutrient absorption is affected, resulting in changes in the development of caryopsis and its substance accumulation.
To my knowledge, these findings are completely new.
The quality of writing is good, the manuscript is clearly written, and the reader can easily follow the story. The manuscript provides new information concerning the relationship between root architecture and the development of caryopsis when barley is subjected to drought stress.

Experimental design

The manuscript is clearly written, the experiments are well planned, and the results are underlined by a sufficient number of replicates.

Validity of the findings

Based on this multidisciplinary approach (morphological and transcriptomic analyses), the authors proposed the model that stress signals from root to caryopsis induced by drought stress induce the differential expression of genes involved in encoding storage proteins and protein biosynthesis pathways, leading to the increase in endosperm protein accumulation in barley caryopsis. The data underlying these results are robust and sound.
Minor revisions:
Figure 3: A picture of a cross section would be helpful to the reader to orientate of which region of the endosperm the close-up was taken. Additionally, the cross section will prove an overview and underline the statistical data and correlate the close-up picture.
Additionally, can the author explain in more detail on the term “relative area” that was mention concerning statistical analyses of protein bodies and of starch granules? It is unclear what the authors finally quantified.
Line 178: The number of vessels in root under DS increased. But the next sentence describes a decrease?

The font and the typing are inconsistent within the legends (missing space after (A), (A) is labelled in bold, ….) The legends should be corrected.

---

## Round 0.2 · accepted · Accept

Based on two reviewers' recommendation, it is acceptable for publication.

Reviewer 1 ·

Basic reporting

/

Experimental design

/

Validity of the findings

/

Additional comments

/

·

Basic reporting

This revised manuscript is good for publication.
Minor changes have been made in attached file.

Experimental design

No comments

Validity of the findings

No comments

Additional comments

Units in Fig. 1 should be put in parenthesis ( ), not / g.

I still think of that DS for drought stress may be not good because the manuscript has DS in many places and also has drought stress in some places. CC is not good for control.